# Maternal BMI During Lactation Is Associated with Major Protein Compositions in Early Mature Milk

**DOI:** 10.3390/nu16223811

**Published:** 2024-11-07

**Authors:** Dong Liang, Zeyu Jiang, Yumei Zhang, Ning Li, Hua Jiang, Gangqiang Ding

**Affiliations:** 1National Institute for Nutrition and Health, Chinese Center for Disease Control and Prevention, Beijing 100050, China; liangdonggrace@outlook.com; 2Department of Nutrition and Food Hygiene, School of Public Health, Peking University, Beijing 100191, China; 1910306135@pku.edu.cn (Z.J.); zhangyumei@bjmu.edu.cn (Y.Z.); 3China National Center for Food Safety Risk Assessment, Beijing 100022, China; lining@cfsa.net.cn; 4School of Nursing, Peking University, Beijing 100191, China; gail_ball@163.com

**Keywords:** human milk, protein, body mass index (BMI), lactoferrin, casein, milk fat globule membrane (MFGM) proteins

## Abstract

Objectives: The present study identified multiple proteins in early mature milk and explored the correlation between protein compositions in HM and maternal BMI during lactation. Methods: A total of 70 mothers giving birth to single-term infants from four representative cites were enrolled in this research. Milk samples were collected between 9 and 11 a.m. to avoid the influence of circadian rhythms. The concentration of total protein in the milk samples was determined using the Bradford method, and the concentrations of α-lactalbumin, lactoferrin, osteopontin, α_s−1_ casein, β-casein, and κ-casein, butyrophilin, periodic acid Schiff 6/7, fatty acid-binding protein, and xanthine oxidoreductase in the milk samples were measured through a previously published method using ultra-performance liquid chromatography coupled with mass spectrometry. A semi-structured questionnaire investigation and body measurements were carried out by trained investigators to collect the information of subjects. Results: In the univariate models, the concentrations of TP (r = 0.306), α-La (r = 0.260), LF (r = 0.371), OPN (r = 0.286), and α_S1_-CN (r = 0.324) were all positively and significantly correlated with maternal BMI. In the models’ adjusted covariates, the concentrations of TP (Lg β = 7.4 × 10^−3^), LF (Lg β = 19.2 × 10^−3^), α_S1_-CN (Lg β = 8.2 × 10^−3^) and the proportion of LF (β = 0.20%) were positively correlated with continuous maternal BMI changes. TP concentrations in the HM of obese mothers were higher than in the other three groups (Lg β: 66.7 × 10^−3^~140.5 × 10^−3^), α-La concentrations were higher than in the underweight and normal groups (Lg β: 94.4 × 10^−3^~145.7 × 10^−3^), and OPN concentrations were higher than in the overweight groups (Lg β = 103.6 × 10^−3^). The concentrations of LF (Lg β: −298.2 × 10^−3^~−191.0 × 10^−3^), OPN (Lg β: −248.9 × 10^−3^~−145.3 × 10^−3^), and α_S1_-CN (Lg β: −160.7 × 10^−3^~−108.3 × 10^−3^) in the HM of underweight mothers were lower than those in the other three groups. β-CN concentrations were lower than normal (Lg β = −125.1 × 10^−3^) and obese groups (Lg β = −165.7 × 10^−3^), κ-CN concentrations were lower than the overweight (Lg β = −132.5 × 10^−3^) and obese groups (Lg β = −147.9 × 10^−3^), and the proportion of LF was lower than that of the overweight (β = −2.80%) and obese groups (β = −2.52%). The proportion of LF in normal mothers was lower than that in the overweight group (β = −1.15%). No statistically significant associations between four MFGM proteins and maternal BMI were determined as the equation models could not be fitted (*p* for F-test < 0.05). Conclusions: Obese mothers had higher concentrations of multiple protein components than other groups, while underweight mothers had lower concentrations. The association between BMI and protein compositions may be more pronounced for certain protein types.

## 1. Introduction

Human milk (HM) is the most comprehensive source of nutrition for infants, containing basic nutrients and a variety of bioactive substances that are beneficial to health [1,2,3]. The World Health Organization (WHO), the United Nations Children’s Fund (UNICEF), and the Chinese Society of Nutrition recommend exclusive breastfeeding (BF) for the first 6 months of life and continued breastfeeding until 2 years old to promote both physical and mental development [4,5,6]. In recent years, many studies have shown the short-term and long-term benefits of BF for both mothers and infants. For lactating women, BF can promote the restoration of the uterus and weight to pre-pregnancy conditions, improve postpartum depression, and reduce the risk of type 2 diabetes, breast cancer, ovarian cancer and many other diseases [1,7,8,9]. For infants, BF can promote cognition and executive function development, protect against infectious and allergic diseases, achieve optimal colonization of the intestinal microbiome, and reduce the risk of obesity and diabetes (both type I and type II) [1,9,10,11,12]. 

Proteins are the third largest macronutrient in HM, and are highly bioavailable [13,14]. There are over 400 proteins contained in HM, which serve as sources of amino acids and essential sources of nitrogen for infants. Furthermore, these nonnutritional bioactive proteins and peptides play a key role in improving the bioavailability of other micronutrients, promoting the development of the infant nervous system and organs, the microbial colonization of the gastrointestinal tract, and the establishment and modulation of immune barriers [3,13,15,16]. HM proteins can be classified into three approximate categories ranging from high to low concentration, as follows: whey proteins, caseins, and milk fat globule membrane (MFGM) proteins [3,15], which account for 60–80%, 20–40%, and 1–4% of the total proteins, respectively [17,18]. The most prominent proteins in whey protein are α-lactalbumin (α-La) and lactoferrin (LF) [15,19]. The former makes up about 22% of the total proteins; this protein is essential for lactose synthesis [20,21], improves intestinal health and immune response in infants, and increases the absorption of mineral such as calcium and zinc [21,22,23]. The latter acts as a transferrin, promotes iron absorption and exerts antibacterial, antiviral, immunomodulatory, and anti-inflammatory effects [19,20,22,23,24,25]. Osteopontin (OPN) has received much attention in recent years, and is only 2% of the total protein. It is associated with biomineralization, tissue remodeling, and immune and metabolism regulation [23,26,27]. Compared to bovine milk, HM casein’s composition makes it easier for infants to digest [28]. Accounting for about 70% of the casein protein, β-casein (β-CN) can facilitate calcium absorption, while κ-casein (κ-CN) possesses antioxidative and antibacterial activities [17,29]. In contrast to α_S2_-casein, found in bovine milk, α_S1_-casein (α_S1_-CN) is the only alpha casein in human milk, and is involved in TLR4-mediated immune responses [30,31]. Among the MFGM proteins, butyrophilin (BTN), periodic acid Schiff 6/7(PAS 6/7), fatty acid-binding protein (FABP), and xanthine oxidoreductase (XOR) occupy a relatively high proportion of the total [18,23]. These proteins have antibacterial and antiviral activity and can help build intestinal immune barriers in infants [23]. 

During the lactation stage, breast milk can be divided into colostrum (0~7 d), transitional milk (8~14 d), early mature milk (15~180 d), and late mature milk (180 d and above). The protein composition and content of colostrum and transitional milk change greatly, while that of mature milk is relatively stable [15,19,32,33,34,35]. In addition, the status of the mothers and the infants, such as nutritional status, delivery mode, baby gender, etc., are important factors affecting the composition of HM proteins.

In recent years, a lot of research has reported that maternal body composition is an important factor affecting the composition of HM [36,37]. Most previous studies focused on the positive correlation between maternal BMI and HM fat concentration [38,39]. To date, only a few studies have focused on the effect on HM proteins. Furthermore, data on some other components (e.g., casein, MFGM proteins) remain scarce.

In the present study, late mature milk with a relatively stable protein content was collected from Chinese lactating women to explore the correlation between the compositions of total protein (TP), whey proteins (α-La, LF, OPN), caseins (α_S1_-CN, β-CN, and κ-CN), and MFGM proteins (BTN, PAS 6/7, FABP, XOR) in HM and maternal BMI during lactation. The results provide suggestions for targeted nutrition and health interventions for lactating women to ensure the optimal development of their infants.

## 2. Materials and Methods

### 2.1. Study Population

This study is part of the 13th Five Year Plan for the National Key Research and Development Program of China (2017YFD0400602). A total of 70 mother–infant dyads from Suzhou (n = 21), Guangzhou (n = 15), Chengdu (n = 19), and Hohhot (n = 15), representing eastern, southern, western, and northern China, were recruited to collect breast milk samples. The eligibility inclusion criteria included (1) lactating mothers between 20 and 45 years; (2) mothers who had given birth 1–3 months ago; (3) singleton and full-term delivery (37–42 weeks); (4) no smoking and alcohol abuse; (5) exclusively breastfed infants. Exclusion criteria included (1) current mastitis or other infectious diseases, (2) major metabolic diseases, (3) no milk sample being collected. The research was conducted in accordance with the Declaration of Helsinki, and the protocol was evaluated and approved by the Medical Ethics Research Board of Peking University (No. IRB00001052-19040). Informed consent for participation was obtained from all respondents involved in the study.

### 2.2. Basic Information Collection and Anthropometric Measurements

Train investigators used semi-structured questionnaires were used to collect the subjects’ information, including maternal age (years), infant age (days), residential city (Suzhou, Guangzhou, Chengdu, Hohhot), delivery modes (cesarean or vaginal), pre-gestational body mass index (BMI, kg/m^2^), gestational weight gain (GWG, kg), infant gender (male, female), and gestational complications (hypertension, diabetes).

A metal column height meter (accurate to 0.1 cm) and double ruler body scale (accurate to 0.1 kg) were used to measure height and weight, respectively. According to the weight criteria for adults, WS/T 428-2013, BMI was calculated as weight (kg)/(height(m))^2^, and categorized into low body weight (BMI < 18.5 kg/m^2^), normal (18.5 kg/m^2^ ≤ BMI < 24.0 kg/m^2^), overweight (24.0 kg/m^2^ ≤ BMI < 28.0 kg/m^2^), and obesity (BMI ≥ 28.0 kg/m^2^) [40]. 

GWG was categorized into inadequate, appropriate and excessive according to the recommendations of the Chinese Nutrition Society (CNS) for weight gain during pregnancy (Table 1) [41]. 

### 2.3. Sample Collection and Preservation

The breast milk sample was collected following the previously established standard process [42]. On the day before the investigation, the investigator contacted the subjects by phone and instructed the mothers to complete breastfeeding and empty their bilateral breasts before 7 a.m. on the day of the investigation. Breast milk samples were collected between 9 and 11 a.m. in the morning to avoid the influence of circadian rhythms. Trained investigators used electric milk pumps to help the maternal subjects to collect a total of 45 mL whole milk (including fore and hind milk) from their unilateral breasts into a sterile tube. The remaining milk was returned to the mother and the milk sample was gently mixed and then immediately stored at −80 °C until testing.

### 2.4. Measurements of Human Milk Protein Concentration

The concentration of TP in the milk samples was determined using the Bradford method [43]. The Coomassie Blue reagent was prepared by dissolving 5 mg of Coomassie Brilliant Blue G 250 in 25 mL of ethanol (95%) and adding 50 mL of phosphoric acid (88%) and 425 mL ultrapure water. This was followed by filtration. Then, 0.1 g milk samples were diluted to 10 mL with water, and 5 µL of diluted milk and 200 µL Coomassie Blue reagent were mixed in a 96-well plate. The percentage of transmittance at 595 nm was recorded after two minutes. The standard curve was generated using bovine serum albumin.

The concentrations of α-La, LF, β-CN, α_S1_-CN, and κ-CN in the milk samples were quantified using the protocol adapted from a previously published method [44]. Each 0.2 g milk sample was diluted to 10 mL with ultrapure water. An aliquot of 10 µL of a diluted milk sample was mixed with 10 µL of an internal standard, 10 µL of dithiothreitol solution (15 mg/mL water), and 845 µL of water. The internal standards were isotope-labeled peptides purchased from ChinaPeptides Co. Ltd. (Shanghai, China). Details of the internal standards are shown in Appendix A. The mixture was then incubated in an 80 °C water bath for 30 min. After that, 10 µL of iodoacetamide solution (54 mg/mL water) was added, and the mixture was left to react at room temperature in the dark for 30 min. For digestion, the mixture was added to 10 µL of trypsin solution (0.4 mg/mL in 1 mmol/L hydrogen chloride) and 100 µL of ammonium bicarbonate solution (39.6 mg/mL in water) and allowed to react in a 37 °C water bath for 4 h. Then, 5 µL of formic acid was added to terminate the digestion. The mixture was homogenized and filtrated through a 0.22 µm nylon filter before further analysis. For quantification, peptide samples were analyzed using acquity ultra-performance liquid chromatography (I-CLASS) coupled with a triple quadrupole mass spectrometer equipped with an electrospray ion source in multiple reaction monitoring modes (Waters, Milford, MA, USA). An Acquity BEH300 C18 column (1.7 µm particle size, 2.1 × 100 mm) was used. Peptides were separated using an 8 min binary gradient consisting of solvent A (water with 0.1% formic acid) and solvent B (acetonitrile with 0.1% formic acid) at a flow rate of 0.3 mL/minute. The elution program started by increasing 3% B to 32% B over 5 min, increasing this to 100% B over 0.1 min, holding at 100% B for 1 min, decreasing to 3% B over 0.1 min, and then holding at 3% B for 1.8 min. The column temperature was 40 °C. The conditions of the mass spectrometer were set as follows: ionization mode, ESI+; capillary voltage, 3.5 kV; source temperature, 150 °C; desolvation temperature, 500 °C; cone gas flow, 150 L/h; desolvation gas flow, 800 L/h; and argon collision gas pressure, 3 × 10^−3^ mbar. The standard curve was generated using protein-specific signature peptides. The concentration of total casein was the sum of the concentrations of β-CN, α_S1_-CN, and κ-CN.

For OPN, BTN, PAS 6/7, BABP, and XOR in the milk samples, each 0.5 g of the milk sample was diluted to 10 mL with ultrapure water. One hundred microliters of the diluted milk sample were then mixed with 50 µL of an internal standard (ChinaPeptides Co. Ltd., Shanghai, China), 780 mL of sodium bicarbonate solution (100 mmol/L), and 10 µL of dithiothreitol solution (500 mmol/L), followed by incubation in a 70 °C water bath for 30 min. After cooling to room temperature, 30 µL of iodoacetamide solution (500 mmol/L) was added to react for 30 min in the dark. For digestion, 20 µL of trypsin solution (1 mg/mL) was added. After overnight digestion at 37 °C, the process was terminated by adding 10 µL of formic acid. After homogenization, the solution was filtrated through a 0.22 µm nylon filter before further analysis. The same ultra-performance liquid chromatography system, column, and mobile phases were used for the determination of OPN; the elution program started by holding solvent B at 5% for 1 min, before increasing it from 5% to 50% over 4.8 min, increasing it again to 100% over 0.2 min, holding at 100% for 0.8 min, decreasing to 5% over 0.2 min, and then holding at 5% for 2 min. The column temperature was 35 °C. Details of the triple quadrupole mass spectrometer conditions are as follows. Multiple reaction monitoring modes were used, as follows: ionization mode, ESI+; capillary voltage, 4.5 kV; capillary temperature, 325 °C; desolvation temperature, 375 °C; desolvation gas flow, 11.5 L/min; and sheath gas flow rate, 10 L/min. For the quantification of MFGM proteins, the elution program started by holding solvent B at 5% for 1 min, increasing it from 5% to 35% over 4 min, increasing it to 100% B over 1 min, holding at 100% B for 2 min, decreasing to 5% B over 0.5 min, and then holding at 5% B for 3 min. The column temperature was 35 °C. Details of the triple quadrupole mass spectrometer conditions were as follows. Multiple reaction monitoring modes were used, as follows: ionization mode, ESI+; capillary voltage, 4.5 kV; capillary temperature, 325 °C; desolvation temperature, 375 °C; desolvation gas flow, 11.5 L/min; and sheath gas flow rate, 10 L/min. The parent ion of BTN was 509.8, while the daughter ions were 497.2 and 426.2; the parent ion of PAS 6/7 was 572.8, while the daughter ions were 532.2 and 445.3; the parent ion of FABP was 454.3, while the daughter ions were 351.2 and 294.3; the parent ion of XOR was 466.7, while the daughter ions were 390.3 and 286.3. All samples were analyzed in duplicate.

The concentrations of HM proteins were expressed as g/100 g. To be consistent with the data in the literature, the concentrations were converted to g/100 mL or mg/100 mL. During the conversion, the HM density was regarded as 103.2 g/100 mL [45]. 

### 2.5. Statistical Analysis

A data analysis was performed using SPSS Statistics (version 27.0.1) and R software (version 4.3.2), and all statistical tests were two-sided, with *p* < 0.05 indicating statistically significant differences. The Kolmogorov–Smirnov test and the Shapiro–Wilk test were applied to confirm whether the distribution of variables obeyed normality. Categorical variables were described as numbers and percentages (n, %); the chi-square test was applied for group comparison. Continuous variables were described as mean ± standard deviation (SD) for normally distributed data or median (P25, P75) for non-normally distributed data. The results of Kolmogorov–Smirnov test and the Shapiro–Wilk test showed that the proportion of each functional protein to the total protein was normally distributed, and the concentration of each functional protein was non-normal before being log-transformed, so the log-transformed value was used for analysis to increase normality. The chi-square test was used to compare the composition ratio between groups, while either Student’s *t*-test or analysis of variance (ANOVA), which was replaced by a nonparametric test when the variable was not normally distributed, were applied to compare the consistency of thee distribution between groups.

The associations between lactation stage and each functional protein were analyzed via Spearman correlation, while the associations between mother’s age, mother’s BMI, and each functional protein were analyzed via Pearson correlation as they were all normally distributed data.

As the correlation coefficient can only indicate whether there is a monotonous linear relationship between variables but cannot control the effect of potential confounding factors, a multiple liner regression analysis was conducted to evaluate the independent effects of maternal BMI on the concentration and proportion of each functional protein compared to the total protein. Model-adjusted covariates included the maternal age (years), lactation stage (days), residential area (west, south, north, east), delivery mode (cesarean or vaginal), GWG (inadequate, appropriate, excessive), infant gender (male, female), and gestational complications (hypertension, diabetes). Model fit was examined using the necessary compliance statistics.

## 3. Results

### 3.1. Subject Characteristics

The general characteristics of the subjects are presented in Table 2. Human milk (HM) samples were collected from 70 mothers aged 22–43 years, The average age of the mothers was 29.7 ± 4.2 years and the average BMI was 23.2 ± 3.4 kg/m^2^. About a third of the mothers had a cesarean section for various reasons, while half of the mothers had approximately appropriate gestational weight gain. Four and seven mothers had records of gestational hypertension and diabetes, respectively. None of the subjects smoked or drank alcohol at any time. Other characteristics of the mothers and infants are shown in Appendix A. All the infants were delivered at full term, including 39 male infants and 31 female infants. The lactation stage ranged from 30 to 136 days (with a median of 51 days). The mothers’ education and economic conditions were investigated, revealing that 74.3% had obtained at least a junior-high school education and had a per capita monthly household income of more than 5000 CNY. In terms of pregnancy and delivery, the rate of first pregnancies (54.3%) and more than one pregnancy (45.7%) was close, while 70% of mothers were primipara. In addition, only one infant was considered to have a growth retardation and none were identified as having a low body weight.

The general characteristics and maternal BMI during lactation are presented in Table 3. No significant differences were found in maternal age and lactation stage among the four BMI groups (*p* > 0.05). Among mothers with an appropriate GWG, a mean BMI of 22.9 ± 2.7 kg/m^2^ was observed; 66.7% of mothers had a normal body weight while 24.2% of them were overweight. It is worth noting that inadequate and excessive GWG were not observed in any subject classified as underweight and obesity, respectively. There were no significant differences in BMI among the four areas of different modes of delivery, infant gender, and gestational complications (*p* > 0.05). On the other hand, mothers who gained excessive weight during gestation had significantly higher BMIs (25.1 ± 3.4 kg/m^2^) than those who gained inadequate (20.8 ± 3.4 kg/m^2^) and appropriate weight (22.9 ± 2.7 kg/m^2^) (*p* < 0.05), and also had higher rates of overweight and obesity than the two groups (*p* < 0.05). The proportion of normal-weight (30.8%) and overweight (23.1%) mothers in the inadequate GWG group was significantly lower than in the other two groups (*p* < 0.05).

### 3.2. Distribution of Breast Milk Proteins

The concentration and proportion of the TP of each protein are presented in Figure 1. The median concentration of total protein in HM was 1.22 (1.06, 1.42) g/100 mL. The proportion of α-La was highest in whey protein, where it is 29.7% of the TP at a concentration of 371.00 (294.48, 459.61) mg/100 mL. The proportion of β-CN is highest in casein, which is 37.9% of the TP at a concentration of 464.27 (402.74, 566.32) mg/100 mL. Whey protein and casein concentrations in breast milk were much higher than in the MFGM protein. The highest concentration in MFGM proteins was XOR, followed by BTN, at 12.40 (9.61, 15.09) mg/100 mL and 6.05 (4.94, 7.66) mg/100 mL, respectively. In general, the concentrations of LF, OPN and β-CN in HM are in accordance with the normal distribution (*p* for Shapiro–Wilk test > 0.05), while the distribution of other proteins conformed to a positive skew distribution due to the influence of outliers (*p* for Shapiro–Wilk test < 0.05, skewness > 0).

### 3.3. Analysis of Total Protein in Human Milk

TP levels in HM according to the BMI category are shown in Table 4. The average Lg TP level was determined to be 0.097 ± 0.088, and it was observed that mothers with obesity had higher TP levels than those who were underweight (*p* < 0.05). TP levels in HM according to other characteristics are shown in Appendix A.

The relationship between maternal BMI and Lg TP levels in the HM is presented in the correlation graph in Figure 2. A moderate positive correlation was identified between Lg TP levels in HM and maternal BMI during lactation (r = 0.306, *p* = 0.010). The relationship between Lg TP levels in the HM and other continuous variables is presented in Appendix A.

The association between maternal BMI and TP levels in HM is presented as a forest plot in Figure 3. The variance inflation factor (VIF) of variables in the models and results of the models diagnostic are presented in Appendix A, which prove that there was no autocorrelation and multicollinearity between the respective variables, and the correlation between maternal BMI and TP levels in HM was significant after adjusting for covariates (*p* for F-test < 0.05). It was determined that Lg TP was independently and positively correlated with maternal BMI changes (β = 7.4 × 10^−3^, *p* < 0.05). Compared to underweight (β = 140.5 × 10^−3^, *p* < 0.05), normal weight (β = 66.7 × 10^−3^, *p* < 0.05), and overweight mothers (β = 67.7 × 10^−3^, *p* < 0.05), those who were obese had higher Lg TP levels in HM.

### 3.4. Analysis of Whey Proteins in Human Milk

The level of each whey protein in HM according to BMI category is shown in Table 5. It was observed that mothers with obesity had higher α-La and OPN levels in HM than those who were underweight (*p* < 0.05). The mothers in the underweight group had lower LF levels in HM than other groups (*p* < 0.05).

The proportion of each whey protein to TP in HM according to BMI category is shown in Table 6. No significant difference was observed in the proportion of three proteins between the four groups (*p* > 0.05). The composition of whey proteins in the HM according to other characteristics is shown in Appendix A.

The relationship between maternal BMI and the levels of each whey protein, as well as their proportion in relation to TP in HM are presented as correlation graphs in Figure 4. A slightly positive correlation was identified between Lg α-La levels (r = 0.260, *p* = 0.030), Lg OPN levels (r = 0.286, *p* = 0.016) in HM, and maternal BMI during lactation. A moderate positive correlation was found between Lg LF levels (r = 0.371, *p* = 0.002) in HM and maternal BMI during lactation. No significant correlation was found between the proportion of the above three whey proteins to TP in HM and maternal BMI (*p* > 0.05). The relationship of the composition of whey proteins in the HM and other continuous variables is presented in Appendix A.

The association between maternal BMI and α-La level and its proportion in relation to TP in HM are presented as a forest plot in Figure 5. The variance inflation factor (VIF) of the variables in models and results of the model diagnostic are presented in Appendix A, which prove that there was no autocorrelation and multicollinearity between respective variables, and the correlation between maternal BMI and α-La level and its proportion in relation to TP in HM is significant after adjusting for covariates (*p* for F-test < 0.05). Compared to underweight (β = 145.7 × 10^−3^, *p* < 0.05) and normal weight mothers (β = 94.4 × 10^−3^, *p* < 0.05), those who were obese had higher Lg α-La levels in HM. It should be noticed that no significant correlation was observed between continuous maternal BMI changes and Lg α-La levels in HM after adjusting for covariates (*p* > 0.05). There is still no significant correlation between the proportion of α-La in relation to TP in HM and maternal BMI (*p* > 0.05).

The association between maternal BMI and LF levels and their proportion in relation to TP in HM are presented as forest a plot in Figure 6. The variance inflation factor (VIF) of the variables in models and results of the model diagnostic are presented in Appendix A, which prove that there was no autocorrelation and multicollinearity between respective variables, and the correlation between maternal BMI and LF level and its proportion to TP in HM is significant after adjusting covariates (*p* for F-test < 0.05). It was determined that Lg LF levels (β = 19.2 × 10^−3^, *p* < 0.05) and its proportions to TP in HM (β = 0.20%, *p* < 0.05) was independently and positively correlated with maternal BMI changes. Compared to underweight mothers, those with normal weight (β = 191.0 × 10^−3^, *p* < 0.05), overweight (β = 262.3 × 10^−3^, *p* < 0.05), and obesity (β = 298.2 × 10^−3^, *p* < 0.05) had higher Lg LF levels in HM. Compared to underweight mothers, those who were overweight (β = 2.80%, *p* < 0.05) and obese (β = 2.52%, *p* < 0.05) had higher LF proportions to TP in HM. Compared to mothers with normal weight, those who were overweight (β = 1.15%, *p* < 0.05) had higher LF proportions to TP in HM.

The association between maternal BMI and OPN level and its proportion in relation to TP in HM are presented as forest plot in Figure 7. The variance inflation factor (VIF) of variables in models and results of the model diagnostic are presented in Appendix A, which prove that there was no autocorrelation and multicollinearity between respective variables. However, only the correlation between maternal BMI category changes and OPN level in HM is significant after adjusting for covariates (*p* for F-test < 0.05). Compared to underweight mothers, those with normal weight (β = 161.6 × 10^−3^, *p* < 0.05), overweight (β = 145.3 × 10^−3^, *p* < 0.05) and obesity (β = 248.9 × 10^−3^, *p* < 0.05) had higher Lg OPN levels in HM. Compared to overweight mothers, those who were obese had higher OPN levels in HM (β = 103.6 × 10^−3^, *p* < 0.05).

### 3.5. Analysis of Casein Proteins in Human Milk

The levels of each casein protein according to BMI category are shown in Table 7. It was observed that underweight mothers had lower β-CN levels in HM than those with normal weight and obesity (*p* < 0.05). Mothers in the underweight group had lower α_S1_-CN levels in HM than the other groups (*p* < 0.05).

The proportions of each casein protein to TP in HM according to BMI category are shown in Table 8. It was observed that underweight mothers had lower α_S1_-CN proportions in relation to TP in HM than normal weight and overweight mothers (*p* < 0.05), and had lower β-CN proportions in relation to TP in HM than mothers of normal weight (*p* < 0.05). The composition of casein proteins in HM according to other characteristics is shown in Appendix A.

The relationship between maternal BMI and each case protein’s level and proportion in relation to TP in HM is presented as correlation graphs in Figure 8. A moderate positive correlation was identified between Lg α_S1_-CN levels (r = 0.324, *p* = 0.006) in HM and maternal BMI during lactation. No significant correlation was found between Lg β-CN, Lg κ-CN, and the proportion of the above three casein proteins in relation to TP in HM and maternal BMI (*p* > 0.05). The relationship between the compositions of casein proteins in the HM and other continuous variables is presented in Appendix A.

The association between maternal BMI and α_S1_-CN levels, and its proportion to TP in HM, are presented as forest plot in Figure 9. The variance inflation factor (VIF) of variables in models and results of the model diagnostic are presented in Appendix A, which prove that there was no autocorrelation and multicollinearity between the respective variables, and the correlation between maternal BMI and α_S1_-CN levels and its proportion in relation to TP in HM is significant after adjusting for covariates (*p* for F-test <0.05). It was determined that Lg α_S1_-CN levels were independently and positively correlated with maternal BMI changes (β = 8.2 × 10^−3^, *p* < 0.05). Compared to underweight mothers, those of normal weight (β = 108.3 × 10^−3^, *p* < 0.05), overweight mothers (β = 125.0 × 10^−3^, *p* < 0.05) and those with obesity (β = 160.7 × 10^−3^, *p* < 0.05) had higher Lg α_S1_-CN levels in HM. There was still no significant correlation between the proportion of α_S1_-CN in relation to TP in HM and maternal BMI (*p* > 0.05).

The association between maternal BMI and β-CN levels, and its proportion in relation to TP in HM, are presented as forest plot in Figure 10. The variance inflation factor (VIF) of the variables in models and results of the model diagnostic are presented in Appendix A, which prove that there was no autocorrelation and multicollinearity between respective variables, and the correlation between maternal BMI and β-CN levels and its proportion to TP in HM was significant after adjusting for covariates (*p* for F-test < 0.05). Compared to underweight mothers, those of normal weight (β = 125.1 × 10^−3^, *p* < 0.05) and those with obesity (β = 165.7 × 10^−3^, *p* < 0.05) had higher Lg β-CN levels in HM. There is still no significant correlation between the proportion of β-CN in relation to TP in HM and maternal BMI (*p* > 0.05).

The association between maternal BMI and Lg κ-CN levels and their proportion in relation to TP in HM are presented as forest plot in Figure 11. The variance inflation factor (VIF) of variables in models and results of the model diagnostic are presented in Appendix A, which prove that there was no autocorrelation and multicollinearity between the respective variables, and the correlation between maternal BMI and κ-CN levels and its proportion in relation to TP in HM is significant after adjusting for covariates (*p* for F-test <0.05). Compared to underweight mothers, those who were overweight (β = 132.5 × 10^−3^, *p* < 0.05) and obese (β = 147.9 × 10^−3^, *p* < 0.05) had higher Lg κ-CN levels in HM. There is still no significant correlation between the proportion of κ-CN in relation to TP in HM and maternal BMI (*p* > 0.05).

### 3.6. Analysis of MFGM Proteins in Human Milk

The levels of each MFGM protein in HM according to BMI category are shown in Table 9. It was observed that underweight mothers had higher BTN and XOR levels in HM than those who were overweight (*p* < 0.05). The composition of MFGM proteins in HM according to other characteristics is shown in Appendix A.

The relationships between maternal BMI and Lg BTN, Lg PAS6/7, Lg FABP, and Lg XOR levels in HM are presented as correlation graphs in Figure 12. No significant correlation was found between the common logarithm levels of the above MFGM proteins in HM and maternal BMI (*p* > 0.05). The relationship between the compositions of MFGM proteins in HM and other continuous variables is presented in Appendix A.

The association between maternal BMI and the level of each MFGM protein in HM is presented as a forest plot in Appendix A. The variance inflation factor (VIF) of variables in the models and results of the model diagnostic are presented in Appendix A, which prove that there was no autocorrelation and multicollinearity between the respective variables, but none of the correlation between maternal BMI and level of MFGM proteins in HM is significant after adjusting covariates (*p* for F-test > 0.05). The effect sizes (*β*) and *p*-values are presented in Appendix A.

## 4. Discussion

In this multicentral cross-sectional study, HM samples were obtained from four representative cities in China, and various functional protein components were determined. We observed a significant positive correlation between maternal BMI during lactation and some protein components of human milk. Mothers with a higher BMI, especially those classified as obese, had significantly higher total protein levels in their milk compared to underweight mothers. This finding highlights the influence of maternal body composition on HM’s nutritional profile, emphasizing BMI’s critical role in determining milk protein content, which is greater than previously recognized. As far as we know, our study is the first study to investigate the relationship between maternal BMI during lactation and to study various HM protein components including MFGM proteins.

In the study population, TP, TF, and κ-CN concentrations in HM were comparable to those reported in previous studies on early mature milk samples from Chinese women. However, our study showed a higher concentration of α-La and β-CN than previous studies, while the concentrations α_S1_-CN and OPN were lower [15,19,46]. Our findings revealed a positive correlation between HM total protein concentration and maternal BMI during lactation, consistent with several existing studies that have explored this relationship. For example, a study involving 80 women reported that total protein concentration in HM was influenced by maternal BMI [47]. Overweight mothers had higher protein concentrations in their breast milk at 5 to 6 months postpartum compared to normal-weight mothers after adjusting for maternal age and infant gender [48]. In the first three months, maternal weight, BMI, and fat mass were positively correlated with the total protein concentration of lactation [36]. 

However, some studies have suggested that the total protein concentration in HM is not directly influenced by BMI, but rather by fat mass. For instance, Kugananthan et al. found that BMI was not associated with the concentration of protein in HM after adjusting for lactation stage, while a higher fat mass percentage was associated with a higher concentration of proteins in HM after accounting for lactation stage [37]. Furthermore, some longitudinal studies and randomized controlled trials also showed no significant difference or correlation in HM protein content between the different BMI groups [39,49,50,51]. These discrepancies suggest that maternal fat mass, rather than BMI, may play a more critical role in determining total protein levels in HM. The variations observed in these studies underscore the complex interplay between maternal BMI, fat mass, and the nutritional makeup of HM. BMI, a widely used matric for assessing body composition, may not fully capture the nuances of fat distribution and muscle mass, which can differ significantly between individuals. This complexity could explain why some studies observed a correlation between BMI and HM protein concentration, while others have not.

One plausible explanation for the differing results is the timing of milk sample collection. Some studies reporting a positive correlation between BMI and HM protein content tended to focus on the later stages of lactation (5–6 months postpartum), while those that did not identify a significant association often conducted sampling during earlier lactation periods (the first three months). This suggests that the relationship between maternal BMI and HM protein content may be dynamic, evolving throughout the lactation process. Early lactation stages are characterized by higher concentrations of immune-regulating proteins, while in mature milk, protein concentrations stabilize. Therefore, differences in lactation stages across studies could contribute to the observed variability in results [52]. 

As for whey proteins, the current study has shown that the concentrations of α-La, LF and OPN were positively correlated with continuous maternal BMI changes before adjusting for covariates. However, the results of the multifactor analysis indicate that only the concentration and proportion of LF to TP are independently correlated with continuous changes in maternal BMI, since the equation model of OPN is not statistically significant. More specifically, mothers with obesity had higher concentrations of both α-La and OPN compared to underweight mothers, which were not accompanied by changes in the proportions of the total protein. A mother-infant dyad study had found similar results, where a higher maternal weight, body mass index, fat-free mass, fat-free mass index, and fat mass index were associated with higher concentration of whey protein [53]. The cohort study of Chen et al. also showed that α-lactalbumin concentration was positively correlated with GWG [54]. 

Some recent studies found that maternal BMI, rather than breastfeeding duration, is positively correlated with children’s obesity indicators. This suggests that maternal body composition may have a direct influence on infant metabolic development [55,56]. Additionally, several studies indicate that the high LF intake of infants is associated with increased weight and fat mass [57]. An animal experiment conducted by Hassan et al. also demonstrated that rats treated stirred yogurt fortified with LF orally for 45 days had higher weight gain and elevated serum LDL and cholesterol levels compared to those fed a basal diet [58]. However, these conclusions are based on bovine LF supplementation in formula, which can not be extended to LF in human milk. Interestingly, in the present study we found that the concentration and proportion of LF to TP in HM are also positively correlated with maternal BMI, being lower in underweight mothers than in other groups. Furthermore, Liu et al. found that, when compared with normal and underweight mothers, the LF content and LF/TP ratio of obese mothers were considerably higher [59]. These findings are in line with our results. Based on this, we propose the following hypothesis: infants’ obesity indicators are influenced by maternal BMI, and this effect is mediated by LF concentration and proportion in HM. However, some studies found there was no significant difference in LF content in HM between mothers of different BMI groups. This discrepancy may be attributed to variations in sample timing and lactation stage [60,61,62]. Therefore, further studies are needed to elucidate the mechanism and association between the above three factors.

Regarding OPN, a similar positive correlation with maternal BMI was found, with obese mothers showing significantly higher OPN concentrations compared to underweight mothers. This result highlights the potential role of maternal BMI in influencing not only the overall protein content but also specific functional proteins such as OPN, which play key roles in infant immune function and development. Consistent with these findings, a previous study has shown that maternal pre-gestational BMI was positively associated with OPN levels at 7 and 14 days postpartum [63]. However, conflicting results have been reported. Some studies have shown a negative correlation between maternal BMI and OPN, particularly when considering GWG. Maternal BMI and GWG were negatively correlated with OPN28. Additionally, some studies found no significant correlation between maternal BMI and milk OPN concentration [64]. 

In the present study, we observed that obese mothers had a higher rate of excessive GWG, while no multicollinearity was detected between GWG and maternal BMI. This could further support the idea that GWG plays a complex role in modulating milk protein composition. Excessive GWG may contribute to a higher BMI and potentially higher α-La and OPN levels in the HM of some mothers.

The mixed results regarding the relationship between maternal BMI, GWG, and OPN levels suggest that this relationship is likely dynamic and context-dependent. In early lactation, a higher pre-gestational BMI may support increased OPN production, as observed in both our study and other studies that focused on the early postpartum period [63]. However, as lactation progresses, excessive weight gain during pregnancy could lead to metabolic stress, potentially inhibiting the continued production of OPN [27]. This dynamic relationship underscores the need for a nuanced understanding of how maternal BMI and weight gain affect milk composition at different stages of lactation. It may also point to the importance of ensuring GWG is within the recommended limits to avoid negative impacts on milk quality, particularly the secretion of critical functional proteins like OPN.

This study shows that maternal BMI is significantly correlated with the concentration of α_S1_-CN and β-CN in HM. α_S1_-CN, which plays a critical role in the efficient transport of casein from the endoplasmic reticulum to the Golgi apparatus, also functions as an important immunomodulator via Toll-like receptor 4 (TLR4) [31,65]. A higher maternal BMI, particularly in obese mothers, may influence α_S1_-CN levels, potentially modifying their functional properties through phosphorylation, and thereby impacting infant immune development.

In addition, β-CN concentrations were positively associated with maternal BMI. Previous research has shown that maternal pre-gestational BMI is correlated with β-CN levels [54], further supporting the notion that mothers with a higher BMI may produce milk that is richer in β-CN-derived peptides, which are vital for maintaining casein micelle stability and promoting gut bioactivity in infants [66]. Although BMI is correlated with the absolute levels of these caseins, there was no significant association between BMI and the proportion of caseins to TP in HM after adjusting for other covariates. This indicates that while maternal BMI impacts the overall concentration of caseins, it has a limited effect on their relative proportion in milk proteins’ composition.

MFGM proteins are an integral part of the milk fat globule membrane structure. In addition to serving as a “container” for milk fat, they also have protective functions. MFGM proteins help to stabilize the milk fat globule and play a role in immune defense by binding to pathogenic microorganisms, preventing them from interacting with the infant’s cells [52]. The MFGM protein content was stable throughout lactation [67]. Compared with proteins, the lipid content of MFGM is relatively variable, influenced by the status of the lactating women and the infants, such as the lactation stage, infant birth weight, gender, and maternal weight [18,68]. 

The current study demonstrates that the concentrations of BTN and XOR are significantly elevated in obese mothers compared to those who are overweight. Although no significant correlation was observed between the continuous variations in maternal BMI and MFGM protein levels, the substantial increase in these proteins among obese mothers suggests that obesity may alter metabolic pathways in the mammary gland, leading to the enhanced synthesis of lipid-associated proteins. Given the critical role of these proteins in infant fat absorption and immune system development, the altered milk composition in obese mothers could have lasting implications for infant health [69]. However, it should be noted that the model fit was not ideal; therefore, the correlation results remain inconclusive and require further studies to confirm. Further research should focus on elucidating the metabolic mechanisms by which maternal obesity modulates MFGM protein levels and assessing the potential long-term effects of these changes on infant growth and development.

In general, the present study found that obese mothers had higher concentrations of multiple protein components than other groups, while low-weight mothers had lower concentrations. Therefore, maintaining a healthy BMI allowed for the optimal composition of multiple proteins in HM, not only promoting healthy growth, the establishment of intestinal flora, and the immune system, but also preventing metabolic disorders such as obesity.

The present study has some limitations. First, in this cross-sectional study, HM was only collected once for each study subject. A correlation between maternal BMI during lactation and the composition of HM proteins was suggested but causality could not be inferred. Second, only early mature milk was collected, and the conclusions cannot be extrapolated to the entire lactation period. Third, due to the limited sampling, the Bradford method was used instead of the Kjeldahl method to determine the concentration of total protein in milk. Fourth, the dietary status of lactation women is very important to the composition of HM proteins [27,70,71,72]. However, only a few subjects were investigated for 24 h and they were not included in the model due to the limited funds and time of the present study. The protein content of mature milk is relatively stable. Except for cases of insufficient and poor-quality dietary protein intake, the protein content is generally not affected by diet factors [73]. Further longitudinal studies or trials should be conducted to clarify the association between the nutritional status and body compositions of lactating women and the composition of HM proteins.

## 5. Conclusions

This study explored the association between maternal BMI during lactation and the compositions of proteins in early mature milk through a cross-sectional study of 70 mother–infant pairs from four representative Chinese cities. The median TP was determined to be 1.22 (1.06, 1.42) g/100 mL, that of α-La was 371.00 (294.48, 459.61) mg/mL, LF was 91.79 (67.01, 111.24) mg/mL, OPN was 19.35 (15.96, 24.44) mg/mL, α_S1_-CN was 52.92 (43.98, 60.07) mg/mL, β-CN was 464.27 (402.74, 566.32) mg/mL, κ-CN was 38.21 (31.6, 49.16) mg/mL, BTN was 6.05 (4.94, 7.66) mg/mL, PAS6/7 was 2.02 (1.55, 2.43) mg/mL, FABP was 0.77 (0.52, 1.16) mg/mL, and XOR was 12.4 (9.61, 15.09) mg/mL. In the univariate models, the concentrations of TP, α-La, LF, OPN, and α_S1_-CN were all positively and significantly correlated with maternal BMI. In the model-adjusted covariates, the concentrations of TP, LF, α_S1_-CN, and the proportion of LF to TP were independently and positively correlated with continuous maternal BMI changes. TP concentrations in the HM of obese mothers were significantly higher than in the other three groups, α -La concentrations were higher than in underweight and normal groups, and OPN concentrations were higher than in overweight groups. The concentrations of LF, OPN, and α_S1_-CN in the HM of underweight mothers were significantly lower than those in the other three groups, β-CN concentrations were lower than in normal and obese groups, κ-CN concentrations were lower than in overweight and obese groups, and the proportion of LF to TP in HM was significantly lower than that of the overweight and obese groups. The proportion of LF to TP in HM was significantly lower than in the overweight group. No statistically significant associations between the four MFGM proteins and maternal BMI were determined as the equation models could not be fitted. Our results underscore the critical role of maternal BMI in shaping the nutritional profile of breast milk, particularly in relation to protein content. This highlights the need for targeted nutritional and health interventions for lactating mothers to ensure optimal infant development. Notably, the study did not find significant associations between maternal BMI and milk fat globule membrane (MFGM) proteins, indicating that the influence of BMI may be more pronounced on certain protein types than others.

## Figures and Tables

**Figure 1 nutrients-16-03811-f001:**
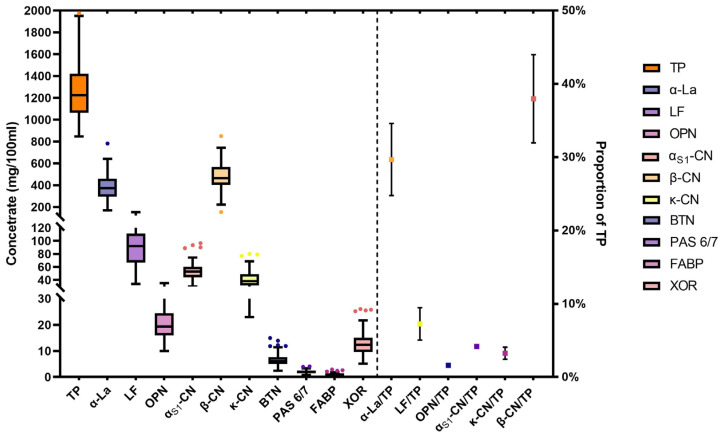
Concentration and proportion of total protein and each functional protein.

**Figure 2 nutrients-16-03811-f002:**
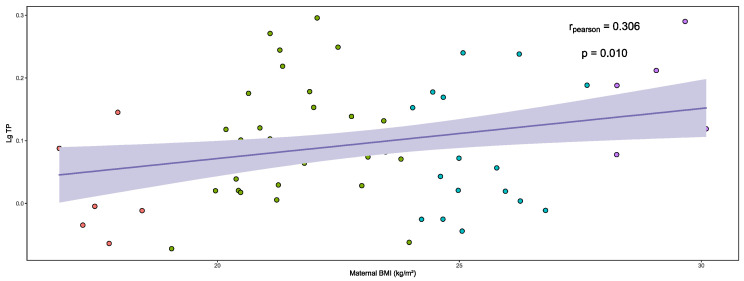
Correlation of Lg TP levels in the HM and maternal BMI. Red: underweight; Green: normal; Blue: overweight; Purple: obesity.

**Figure 3 nutrients-16-03811-f003:**
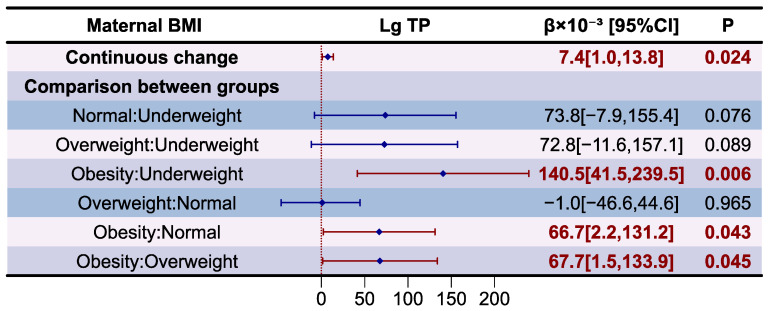
Associations between maternal BMI and HM total proteins according to multiple linear regression analysis. The model was adjusted according to lactation stage (days), maternal age (years), residential area (west, south, north, east), delivery mode (cesarean, eutocia), GWG (inadequate, appropriate, excessive), infant gender (male, female), and gestational complications (hypertension, diabetes).

**Figure 4 nutrients-16-03811-f004:**
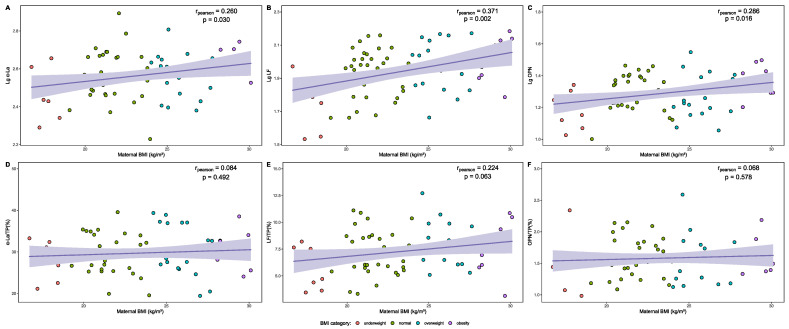
Correlation of composition of whey proteins in the HM and maternal BMI. (**A**) Correlation between maternal BMI and Lg α-La in the HM. (**B**) Correlation between maternal BMI and Lg LF in the HM. (**C**) Correlation between maternal BMI and Lg OPN in the HM. (**D**) Correlation between maternal BMI and α-La/TP (%) in the HM. (**E**) Correlation between maternal BMI and LF/TP (%) in the HM. (**F**) Correlation between maternal BMI and OPN/TP (%) in the HM.

**Figure 5 nutrients-16-03811-f005:**
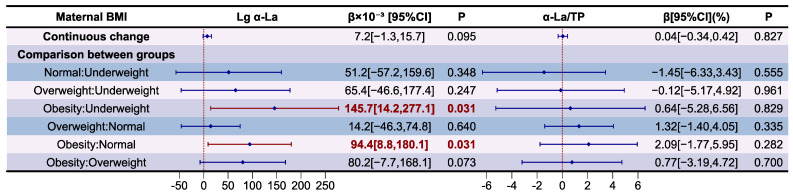
Associations between maternal BMI and α-La according to multiple linear regression analyses. The model was adjusted according to lactation stage (days), maternal age (years), residential area (west, south, north, east), delivery mode (cesarean, eutocia), GWG (inadequate, appropriate, excessive), infant gender (male, female), and gestational complications (hypertension, diabetes).

**Figure 6 nutrients-16-03811-f006:**
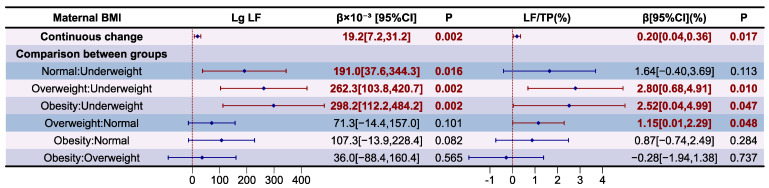
Associations between maternal BMI and LF according to multiple linear regression analyses. The model was adjusted according to lactation stage (days), maternal age (years), residential area (west, south, north, east), delivery mode (cesarean, eutocia), GWG (inadequate, appropriate, excessive), infant gender (male, female), and gestational complications (hypertension, diabetes).

**Figure 7 nutrients-16-03811-f007:**
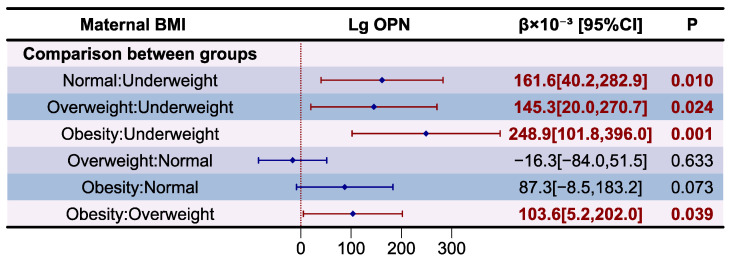
Associations between maternal BMI and OPN according to multiple linear regression analyses. The model was adjusted according to lactation stage (days), maternal age (years), residential area (west, south, north, east), delivery mode (cesarean, eutocia), GWG (inadequate, appropriate, excessive), infant gender (male, female), and gestational complications (hypertension, diabetes).

**Figure 8 nutrients-16-03811-f008:**
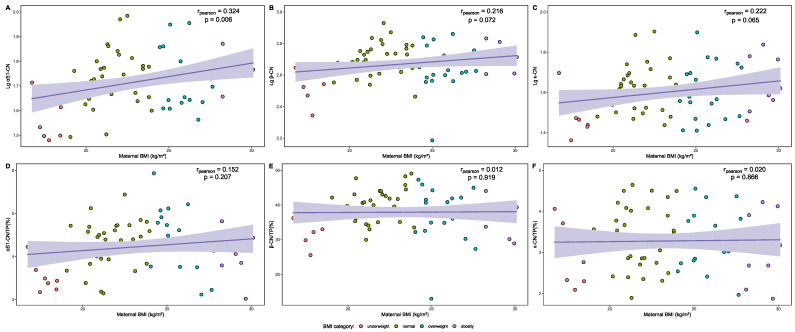
Correlation of composition of casein proteins in the HM and maternal BMI. (**A**) Correlation between maternal BMI and Lg α_S1_-CN in the HM. (**B**) Correlation between maternal BMI and Lg β-CN in the HM. (**C**) Correlation between maternal BMI and Lg κ-CN in the HM. (**D**) Correlation between maternal BMI and α_S1_-CN/TP (%) in the HM. (**E**) Correlation between maternal BMI and β-CN/TP (%) in the HM. (**F**) Correlation between maternal BMI and κ-CN/TP (%) in the HM.

**Figure 9 nutrients-16-03811-f009:**
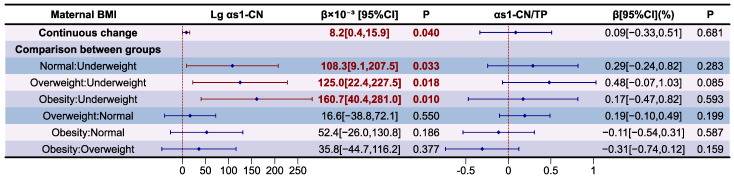
Associations between maternal BMI and α_S1_-CN according to multiple linear regression analyses. The model was adjusted according to lactation stage (days), maternal age (years), residential area (west, south, north, east), delivery mode (cesarean, eutocia), GWG (inadequate, appropriate, excessive), infant gender (male, female), and gestational complications (hypertension, diabetes).

**Figure 10 nutrients-16-03811-f010:**
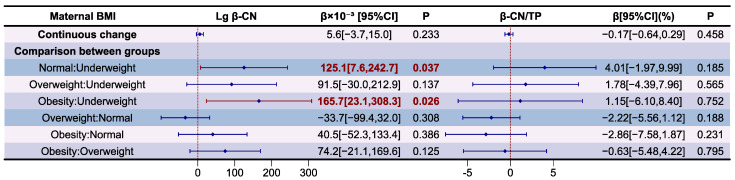
Associations between maternal BMI and β-CN according to multiple linear regression analyses. The model was adjusted according to lactation stage (days), maternal age (years), residential area (west, south, north, east), delivery mode (cesarean, eutocia), GWG (inadequate, appropriate, excessive), infant gender (male, female), and gestational complications (hypertension, diabetes).

**Figure 11 nutrients-16-03811-f011:**
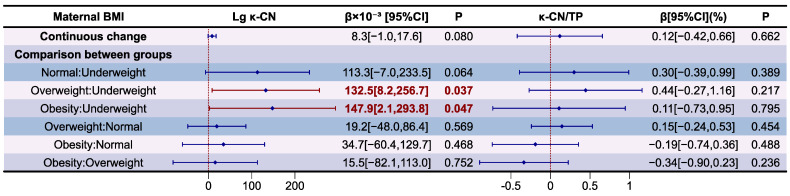
Associations between maternal BMI and κ-CN according to multiple linear regression analyses. The model was adjusted according to lactation stage (days), maternal age (years), residential area (west, south, north, east), delivery mode (cesarean, eutocia), GWG (inadequate, appropriate, excessive), infant gender (male, female), and gestational complications (hypertension, diabetes).

**Figure 12 nutrients-16-03811-f012:**
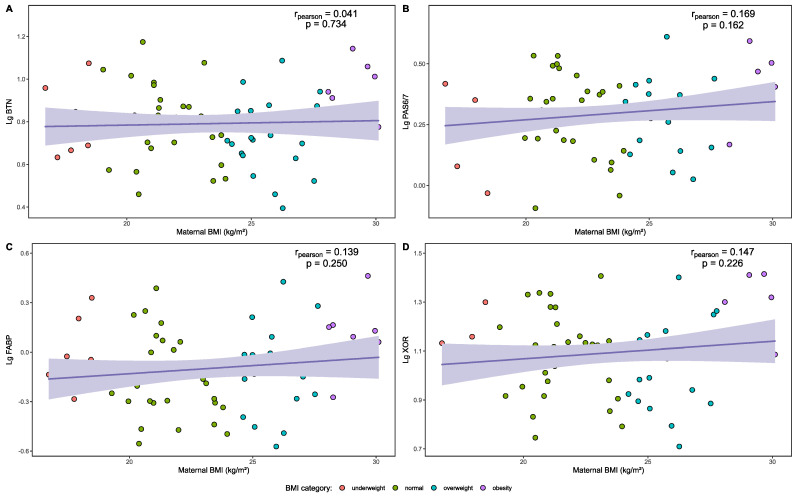
Correlation of common logarithm levels of MFGM proteins in the HM and maternal BMI. (**A**) Correlation between maternal BMI and Lg BTN in the HM. (**B**) Correlation between maternal BMI and Lg PAS 6/7 in the HM. (**C**) Correlation between maternal BMI and Lg FABP in the HM. (**D**) Correlation between maternal BMI and Lg XOR in the HM.

**Table 1 nutrients-16-03811-t001:** Range of weight gain in pregnant women according to BMI classification before pregnancy (CNS).

Pre-Gestational BMI	Category of Gestational Weight Gain		
	Inadequate	Appropriate	Excessive
Underweight (BMI < 18.5 kg/m^2^)	GWG < 11 kg	11 ≤ GWG ≤ 16	GWG > 16 kg
Normal (18.5 kg/m^2^ ≤ BMI < 24.0 kg/m^2^)	GWG < 8 kg	8 ≤ GWG ≤ 14	GWG > 14 kg
Overweight (24.0 kg/m^2^ ≤ BMI < 28.0 kg/m^2^)	GWG < 7 kg	7 ≤ GWG ≤ 11	GWG > 11 kg
Obesity (BMI ≥ 28.0 kg/m^2^)	GWG < 5 kg	5 ≤ GWG ≤ 9	GWG > 9 kg

GWG: gestational weight gain.

**Table 2 nutrients-16-03811-t002:** General characteristics of subjects.

Variables	Descriptive Value
Maternal age (year, $\bar{x}\pm s$)	29.7 ± 4.2
Lactation stage (day, M(P25, P75))	51 (37, 71)
Maternal BMI (kg/m^2^, $\bar{x}\pm s$)	23.2 ± 3.4
Area (%)	
West	19 (27.1%)
South	15 (21.4%)
North	15 (21.4%)
East	21 (30.0%)
Cesarean (%)	25 (35.7%)
GWG (%)	
Inadequate	13 (18.6%)
Appropriate	33 (47.1%)
Excessive	24 (34.3%)
Infant gender (%)	
Male	39 (55.7%)
Female	31 (44.3%)
Gestational hypertension (%)	4 (5.7%)
Gestational diabetes (%)	7 (10.0%)

**Table 3 nutrients-16-03811-t003:** General characteristics and maternal BMI during lactation.

Variables	Maternal BMI (kg/m^2^, $\bar{x}\pm s$)	Underweight	Normal	Overweight	Obesity	P_1_	P_2_
Maternal age (year, $\bar{x}\pm s$)	-	27.7 ± 3.9	29.5 ± 4.4	31.2 ± 4.2	28.8 ± 2.8	0.202	
Lactation stage (day, M(P_25_, P_75_))	-	65 (50, 87)	51 (35, 74)	52 (35, 68)	39 (34, 64)	0.220	
Area (%)						0.135	0.589
West	22.6 ± 3.4	2 (10.5)	11 (57.9)	4 (21.1)	2 (10.5)		
South	22.3 ± 3.8	4 (26.7)	6 (40.0)	4 (26.7)	1 (6.7)		
North	24.9 ± 2.9	0	7 (46.7)	5 (33.3)	3 (20.0)		
East	23.3 ± 3.2	1 (4.8)	11 (52.4)	7 (33.3)	2 (9.5)		
Delivery mode (%)						0.703	0.979
Cesarean	23.4 ± 3.4	3 (12.0)	12 (48.0)	7 (28.0)	3 (12.0)		
Eutocia	23.1 ± 3.4	4 (8.9)	23 (51.1)	13 (28.9)	5 (11.1)		
GWG (%)						<0.001	<0.001
Inadequate	20.8 ± 3.4 ^a^	6 (46.2) ^a^	4 (30.8) ^b^	3 (23.1) ^b^	0 ^a^		
Appropriate	22.9 ± 2.7 ^a^	1 (3.0) ^a^	22 (66.7) ^a^	8 (24.2) ^a^	2 (6.1) ^a^		
Excessive	25.1 ± 3.4 ^b^	0 ^a^	9 (37.5) ^a^	9 (37.5) ^ab^	6 (25.0) ^b^		
Infant gender (%)						0.646	0.691
Male	23.4 ± 3.5	4 (10.3)	17 (43.6)	13 (33.3)	5 (12.8)		
Female	23.0 ± 3.3	3 (9.7)	18 (58.1)	7 (22.6)	3 (9.7)		
Gestational hypertension (%)						0.615	0.481
Yes	23.0 ± 0.8	0	4 (100.0)	0	0		
No	23.2 ± 3.5	7 (10.5)	31 (47.0)	20 (30.3)	8 (12.1)		
Gestational diabetes (%)						0.705	0.617
Yes	22.8 ± 2.7	1 (14.3)	3 (42.9)	3 (42.9)	0		
No	23.3 ± 3.5	6 (9.5)	32 (50.8)	17 (27.0)	8 (12.7)		

P_1_: Student *t*-test, ANOVA test or Kruskal–Wallis test; P_2_: χ^2^ test or Fisher test. a,b: A statistically significant difference between binary groups is expressed using the Bonferroni method.

**Table 4 nutrients-16-03811-t004:** TP levels in the HM according to BMI category.

BMI	Lg TP	*p*
Overall	0.097 ± 0.088	0.021
Underweight	0.028 ± 0.076 ^a^	
Normal	0.100 ± 0.085 ^a,b^	
Overweight	0.090 ± 0.090 ^a,b^	
Obesity	0.167 ± 0.065 ^b^	

*p*: ANOVA test; a,b: A statistically significant difference between binary groups is expressed using the Bonferroni method.

**Table 5 nutrients-16-03811-t005:** Whey proteins levels in HM according to BMI category.

BMI	Lg α-La	*p*	Lg LF	*p*	Lg OPN	*p*
Overall	2.564 ± 0.123	0.036	1.939 ± 0.156	0.003	1.287 ± 0.122	0.018
Underweight	2.467 ± 0.133 ^a^		1.755 ± 0.163 ^a^		1.180 ± 0.120 ^a^	
Normal	2.562 ± 0.127 ^a,b^		1.935 ± 0.140 ^b^		1.296 ± 0.107 ^a,b^	
Overweight	2.566 ± 0.110 ^a,b^		1.979 ± 0.144 ^b^		1.276 ± 0.130 ^a,b^	
Obesity	2.651 ± 0.074 ^b^		2.016 ± 0.142 ^b^		1.372 ± 0.104 ^b^	

*p*: ANOVA test; a,b: A statistically significant difference between binary groups is expressed via the Bonferroni method.

**Table 6 nutrients-16-03811-t006:** Proportions of whey proteins to TP in the HM according to BMI category.

BMI	α-La/TP(%)	*p*	LF/TP(%)	*p*	OPN/TP(%)	*p*
Overall	29.7 ± 4.9	0.555	7.3 ± 2.1	0.091	1.6 ± 0.3	0.786
Underweight	27.8 ± 4.7		5.7 ± 2.1		1.5 ± 0.4	
Normal	29.4 ± 4.4		7.1 ± 2.0		1.6 ± 0.3	
Overweight	30.5 ± 5.9		8.0 ± 1.9		1.6 ± 0.4	
Obesity	30.8 ± 4.7		7.5 ± 2.6		1.6 ± 0.3	

*p*: ANOVA test.

**Table 7 nutrients-16-03811-t007:** Casein proteins levels in the HM according to BMI category.

BMI	Lg α_S1_-CN	*p*	Lg β-CN	*p*	Lg κ-CN	*p*
Overall	1.719 ± 0.113	0.002	2.670 ± 0.122	0.006	1.601 ± 0.125	0.064
Underweight	1.573 ± 0.094 ^a^		2.539 ± 0.109 ^a^		1.486 ± 0.107	
Normal	1.730 ± 0.107 ^b^		2.696 ± 0.098 ^b^		1.615 ± 0.110	
Overweight	1.728 ± 0.112 ^b^		2.648 ± 0.145 ^a,b^		1.603 ± 0.138	
Obesity	1.775 ± 0.063 ^b^		2.724 ± 0.083 ^b^		1.639 ± 0.138	

*p*: ANOVA test; a,b: a statistically significant difference between binary groups is expressed using the Bonferroni method.

**Table 8 nutrients-16-03811-t008:** Proportions of casein proteins to TP in the HM according to BMI category.

BMI	α_S1_-CN/TP(%)	*p*	β-CN/TP(%)	*p*	κ-CN/TP(%)	*p*
Overall	4.2 ± 0.6	0.005	37.9 ± 6.0	0.023	3.3 ± 0.7	0.524
Underweight	3.5 ± 0.4 ^a^		32.7 ± 4.0 ^a^		3.0 ± 0.8	
Normal	4.3 ± 0.5 ^b^		39.7 ± 4.6 ^b^		3.4 ± 0.7	
Overweight	4.4 ± 0.7 ^b^		37.2 ± 7.9 ^a,b^		3.3 ± 0.7	
Obesity	4.1 ± 0.6 ^a,b^		36.4 ± 5.1 ^a,b^		3.1 ± 0.9	

*p*: ANOVA test; a,b: A statistically significant difference between binary groups is expressed using the Bonferroni method.

**Table 9 nutrients-16-03811-t009:** MFGM proteins levels in the HM according to BMI category.

BMI	Lg BTN	*p*	Lg PAS 6/7	*p*	Lg FABP	*p*	Lg XOR	*p*
Overall	0.791 ± 0.170	0.043	0.294 ± 0.149	0.205	−0.099 ± 0.239	0.060	1.092 ± 0.165	0.032
Underweight	0.801 ± 0.165 ^a,b^		0.232 ± 0.157		−0.028 ± 0.225		1.091 ± 0.113 ^a,b^	
Normal	0.792 ± 0.160 ^a,b^		0.286 ± 0.151		−0.149 ± 0.221		1.084 ± 0.160 ^a,b^	
Overweight	0.730 ± 0.177 ^a^		0.292 ± 0.143		−0.110 ± 0.256		1.045 ± 0.172 ^a^	
Obesity	0.930 ± 0.138 ^b^		0.389 ± 0.133		0.091 ± 0.209		1.244 ± 0.135 ^b^	

*p*: ANOVA test; a,b: A statistically significant difference between binary groups is expressed using the Bonferroni method.

## Data Availability

The datasets analyzed in this study are available from the corresponding author upon reasonable request due to privacy policy.

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
