# Peer review of "Maternal BMI During Lactation Is Associated with Major Protein Compositions in Early Mature Milk"

_nutrients, 2024, doi:10.3390/nu16223811_

Round 1

Reviewer 1 Report

Comments and Suggestions for Authors

An interesting and descriptive article on the levels of functional proteins in healthy Chinese women in mature milk. In general, it is very well-described and replicable work. However, there are multiple typographical errors of spaces, capital letters, etc. (Lines: 196, 202, 210,215, 217, 228, 239, 240, 497, 558, Figure 1 in Y-axis "concentration or level "...).

- It would be necessary to summarize many of the texts as the abstract or the discussion, because the focus of the work is lost. Furthermore, it is necessary to implement conclusions, perhaps some of the "patent" or discussion text can be moved to the conclusions.

- Women with obesity/overweight had higher levels of these functional proteins. Is maternal obesity better to pass more level of these proteins to the neonate? How is it related to the deposits that the woman has stored during her pregnancy? Is there any association of these levels with the GWG?

-Maternal diet (DOI: 10.3389/fendo.2023.1090499) and neonatal sex (DOI: 10.3390/antiox11081472) strongly influence the composition of human milk, especially functional proteins. How are these data related to?

Some comments to discuss with the authors could be:

- Section 2.4. Why were the Pearson and Spearman correlations used? It would be convenient to use the Spearman correlations since it is restrictive with the sample size. It would be usefull to report which model statistics fit were used.

- Section 3.2. Are the protein levels detected low?

- Figure 1 and Table 4 report the same data, one of them should be chosen.

- Correlations could occur throughout the text and ignore the scatterplot.

- Figure 13 is difficult to see.

Author Response

Thanks for your review and suggestion.We have revised our manuscript and submitted to the responsible editor.Revises and comments of the manuscript have been marked using revision mode of word document.Below I will respond to your valuable suggestions:

1.I am sure of some typographical mistakes occurred so that I can’t find the errors you raised in the corresponding line.So we checked and amended some texts and figures.

2.Some texts of “patent” and discussion have been moved to the conclusion.

3.The relationship between breast milk protein and infant growth and development was analyzed and discussed in another article of this research group (DOI: 10.3390/nu13051476). Therefore, this problem is not covered in this study.

4.Since univariate analysis showed that the BMI of the subjects in different GWG groups was not exactly the same (Table 3,P<0.001), we suspected that there was a collinearity between the two variables.However, the diagnostic results of multiple linear regression (Table S7) showed that neither of them had collinearity (VIF<10).Besides, the results of univariate analysis between GWG and each protein component are shown in the Table S2~S5. Finally, only part of the protein compositions were independently related to GWG according to the results of multiple linear regression, we did not discuss them separately due to length and theme of the article.

5.In line 678 of the revised manuscript we mentioned the reason why we did not discuss the mothers’ diet.

6.Our t-test results did not reveal significant differences in each protein composition in HM between different infant gender (Table S2~S5, P>0.05).That’s why we did not perform a subgroup analysis for different infant gender.

7.The conditions of use for Pearson correlation or Spearman correlation analysis have been added to the statistical analysis section.We made a selection based on the normality of the data.

8.In line 678 of the revised manuscript we converted and compared the protein levels measured in this study with those in some other studies.

9.In Figure 1 and Table 4, we modified and retained Figure 1.

10.Since none of the models shown in Figure 13 were statistically significant (P for F-test> 0.05), we moved them to supplemental material and used a higher definition picture.

Reviewer 2 Report

Comments and Suggestions for Authors

I have read this paper with great interest and do value the effort and the paper as currently submitted, In essence, I have only specific comments and reflections

First, while the correlations are indeed statistically significant, these correlations likely only marginally explain the observed variability. I draw this conclusion based on the different figures provided by the authors (like figure 2, figure 4, figure 8, figure 12), but there would be add on value if the authors can at least further stress this.

In my opinion, it should be clearly provided when (days postpartum) the samples were collected in the different weight categories, to ensure that this is indeed not significantly different between subgroups.

We do need some more details on the milk collection: one breast, or both, and full pumped with subsequent collection of 45 ml ? and how was milk collected (expression, electric or mechanical pump).

Editing

Line 39: not sure why you use low-weight mothers overhere, suggest to use consistent wording throughout the paper.

Line 262: what do you mean with ‘than those two groups ?’

Lines 664-667: I assume something went wrong overhere, and that ‘this section…’ and patents should be removed.

Author Response

Thanks for your review and suggestion.We have revised our manuscript and submitted to the responsible editor.Revises and comments of the manuscript have been marked using revision mode of word document.Below I will respond to your valuable suggestions:

1.We highlighted the limitations of the correlation analysis you mentioned in the statistical analysis section and clarified why we should go further with multiple linear regression.

2.At your suggestion, we have added a comparison of lactation stage and nursing age between different BMI groups in Table 2.We agree that this is really necessary.

3.We have added the details of breast collection as you suggested.

4.I am sure of some typographical mistakes occurred, so we checked and amended some texts and figures.The presentation errors you raised have been revised. And some texts of “patent” and discussion have been moved to the conclusion.